# Emulating Oncologists' Gaze for Predicting Treatment Response through Multimodal Imaging

## Abstract

We used three datasets to imitate oncologists' decisions about the prognostic response by looking at lung tumors in different imaging modalities, i.e., PET and CT. We extract comprehensive visual features, *radiomics*, from the tumors' region and then reduce their size with a density-based isometric mapping while preserving their main visual characteristics. We apply the Parzen-Rosenblatt (PR) constrain to modify isometric mapping. For the comparison, we use two metrics, binary classification and Cox proportional hazard models, to avoid biases in the comparison. We achieved prediction accuracy comparable to newly and commonly established methods. We successfully predict patient outcomes in response to therapy and imitate the oncologist's attention over multimodal images.

## 1. Introduction

Radiologic imaging features, *radiomics*. are used to train machine learning algorithms and predict responses to specific treatments for cancer patients. The abundance of radiomics, high-dimensional (HD), impedes the models to accurately predict the outcome. Projecting HD data in lower dimension (LD) space using methods like isometric embedding Tenenbaum et al. (2000) is always interesting since the data becomes sparse by the soar of dimension while preserving the overall characteristics of the HD data Jolliffe (2011); Torgerson (1965); Moon et al. (2019); Xu et al. (2022); Yang et al. (2022). We propose density-based constrain to control isometric embedding resulting in prediction response to therapy by looking at multimodal imaging. We use three non-small cell lung cancer (NSCLC) datasets, NSCLC Radiogenomics Bakr et al. (2017); Gevaert et al. (2012); Clark et al. (2013), NSCLC-Radiomics (LUNG01) Aerts et al. (2017, 2014), and NSCLC-Radiomics (LUNG03) Aerts et al. (2017, 2014), which are focus on computed tomography (CT) and positron emission tomography (PET) to build therapy response prediction and compare it with actual results, as oncologists' gaze estimation. Figures 1,2,3,4 presents the overall strategy of the proposes system, examples of multimodal radiologic imaging, how visual data is reduced to LD space without any collinearity, and the final treatment response prediction of models with the given LD data, respectively. Whereas an oncologist and a radiologist look at visual features and may encounter missing or overwhelming information, data reduction methods are expected to highlight important features, which causes a considerable impact on patient outcome predictions.

Isometric mapping, unlike classical multidimensional scaling (MDS) Torgerson (1965), utilizes geodesic distance to calculate distances between points on a curved manifold, which is advantageous for distant points. However, standard isometric mapping relies on the k-nearest neighbor approach to find the shortest path Tenenbaum et al. (2000), which may not guarantee close neighbor proximity. When a large k is selected, points that are sufficiently far from each other on the manifold may lead to an approximation of geodesic distances with Euclidean distances, similar to the issue encountered in MDS.

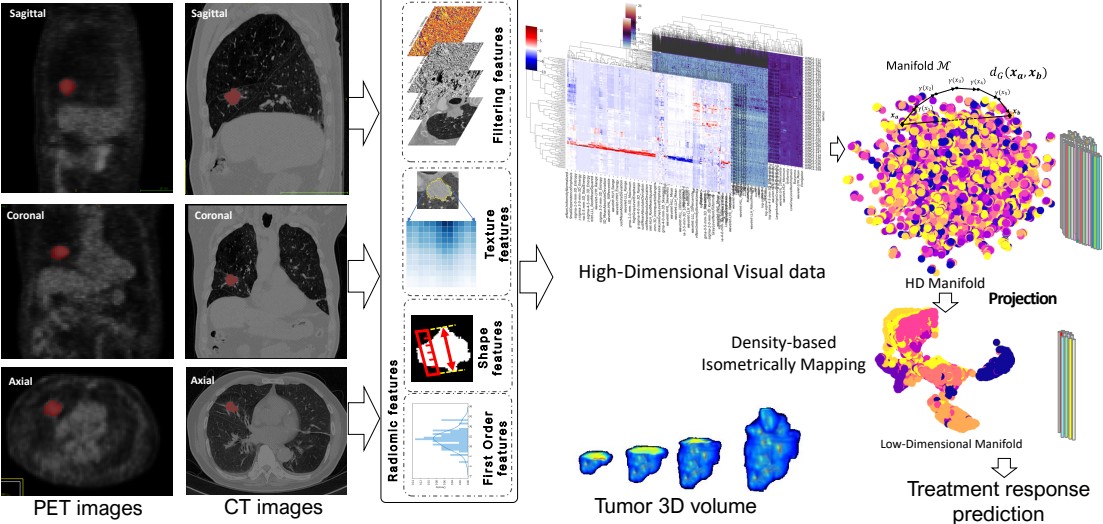

Figure 1: Our density-based isometric mapping deflates an HD manifold onto an LD representation while preserving their similarities. PR-Isomap preserves similarity in LD embedding and showed more promising results in predicting patients' survival.

To relax this and enforce the geodesic points to stay on the manifold we add a Parzen – Rosenblatt (PR) Parzen (1962); Rosenblatt (1956) constrain to modify local neighbors' graph to highlight the uniformity of the data distribution on the HD manifold for isometric mapping, density-based isometric mapping, which preserves both distances and uniformity criteria while nonlinearly reducing the dimensionality. Given the application of the shortest path, we modify Dijkstra's algorithm, as shown in Appendix B, with the computational complexity of the algorithm, Appendix D. Finally, we predict response to therapy using binary classification and Cox proportional hazard (CPH) models reaching comparable accuracies, which is confirmed by Kaplan Meier survival curves, Figure 4.

## 2. Related Work

In the field of cancer prediction response, few works have used radiological feature reduction methods to predict treatment response Limkin et al. (2017); Yousefi et al. (2021); Hosny et al. (2019); Chetan and Gleeson (2021); He et al. (2020). However, only a few works have applied isometric mapping to preserve the visual information from radiologic images Limkin et al. (2017); Peng et al. (2018). Peng et al. (2018) Peng et al. (2018) used this method to distinguish true progression from radionecrosis after radiation therapy for metastatic brain cancer in magnetic resonance imaging (MRI). Besides isometric embedding, different data reduction approaches have been used for NSCLC prognosis, such as principal component analysis PCA Horng et al. (2022), t-distribution stochastically embedding (t-SNE) Van der Maaten and Hinton (2008); Ben-Hamo et al. (2020); Chen et al. (2023), and Potential of Heat-diffusion for Affinity-based Trajectory Embedding (PHATE) Moon et al. (2019);

Wiecek et al. (2023). To the best of our knowledge, this is the first study to use density-based isometric mapping to predict treatment response on multimodal imaging.

## 3. Methods

In the following sections, we introduce the isometric mapping method and discuss the innovative techniques used to extract maximum LD information for predicting survival and classification tasks (Figure 1).

### 3.1. Density isometric mapping

Our data points can be shown as a combination of multiple vectorized data points, $X = [\mathbf{x}_1, \mathbf{x}_2, \ldots, \mathbf{x}_n] \in \mathbb{R}^{d \times n}$, where $\left\{\mathbf{x}_i \in \mathbb{R}^d\right\}_{i=1}^{n}$. We assume that there is a finite data point drawn from our X. We can build a geometric graph $G = (V; E)$ that has our data points as vertices and connects vertices that are close to each other. We construct a k-nearest neighbor graph for the $\mathbf{x}_i$ connected to $\mathbf{x}_j$ correspondingly and for $\mathbf{x}_j$ to $\mathbf{x}_i$, show them by $(\mathbf{x}_i)$ and $\gamma(\mathbf{x}_j), \gamma(\mathbf{x}_i), \gamma(\mathbf{x}_j) \in \mathcal{M}$. $\gamma(.)$ represents a vector point on the manifold. The length of the path is defined by summing the edge weights along the path on the manifold. The shortest path $(\mathcal{SP})$ distance $D_{\mathcal{SP}}(\gamma(\mathbf{x}_i), \gamma(\mathbf{x}_j))$ between two vertices $\gamma(\mathbf{x}_i), \gamma(\mathbf{x}_j)$ and defines based on the length of the $\mathcal{SP}$ connecting them. $g(\cdot)$ is a positive continuous function for X in the given path $\gamma(\cdot)$ connecting $\mathbf{x}_i$ and $\mathbf{x}_j$, which is parameterized by $g$-length of the path as

$$D_{g,\gamma}(t) = \int_{\gamma} g(\gamma(t)) \mid \gamma'(t) \quad dt, \tag{1}$$

where $\gamma'(\cdot)$ defines a small element on the manifold, also known as the line integral along the path $\gamma(\cdot)$ with respect to $g(\cdot)$. $D_{g,\gamma}$ denotes as distance on the manifold through the shortest path.

Geodesic distance of $\gamma(\mathbf{x}_i)$ and $\gamma(\mathbf{x}_j), (\gamma(\mathbf{x}_i), \gamma(\mathbf{x}_j)) \in \mathcal{M}$ defines along the manifold surface, $\mathcal{M}$, and is denoted by smaller steps Euclidean distances $D_{Euc}(\mathbf{x}_i, \mathbf{x}_j)$ and can be approximate by $D_{g,\gamma}(\mathbf{x}_i, \mathbf{x}_j)$.

By building a neighborhood graph where each point is connected exclusively to its k nearest neighbors, Isomap first calculates an estimate of the geodesic distances between every pair of data points located on the manifold $\mathcal{M}$; the edge weights are equivalent to the corresponding pairwise distances. The Euclidean distance serves as an estimate for the geodesic distance for adjacent pairs of data points. This assumption allows finding an approximation of $D_{g,\gamma}(\mathbf{x}_i, \mathbf{x}_j)$ applying many small Euclidean distances, e.g.,

$$D_{g,\gamma}(\mathbf{x}_i, \mathbf{x}_j) \approx \|\gamma(\mathbf{x}_i) - \gamma(\mathbf{x}_j)\|_2 \quad \forall \quad \gamma(\mathbf{x}_j) \in \mathcal{N}_k(\gamma(\mathbf{x}_j)), \tag{2}$$

where $\mathcal{N}_k(\gamma(\mathbf{x}_j))$ denotes the collection of the point $\gamma(\mathbf{x}_j) \in \mathcal{M}$ for $k$ closest neighbors on the manifold surface, $\mathcal{M}$. The geodesic distance is calculated for non-neighboring points as the shortest path length along the neighborhood graph, which can be discovered using Dijkstra's algorithm Dijkstra (2022). The generated geodesic distance matrix is then subjected to MDS Torgerson (1965) to identify a group of LD points that most closely match such distances, more details are in Appendix A. Creating a k-nearest neighbor graph for

data points can introduce inconsistency between geodesic and Euclidean distances, especially when the third Cayton assumption Cayton (2005); Babaeian (2017) is not strongly satisfied. This assumption implies a uniform distribution of points in the high-dimensional manifold. However, with a large $k$ in the k-nearest neighbor graph, discrepancies can arise in approximating geodesic and Euclidean distances, leading to differences between intrinsic and extrinsic distances on the manifold.

Here, we tackle this problem using the PR window constrain Parzen (1962); Rosenblatt (1956) on the k-nearest neighbor for a weak-uniformly distributed manifold which enforces the accuracy of distance approximation in the isometric mapping algorithm (Algorithm 1 and Fig. **??**). In this respect,

**Definition 1 (PR density window)** *Parzen–Rosenblatt (PR) window is defined as:*

$$p_h\left(\mathbf{x}\right) = \frac{1}{k} \sum_{i=1}^{k} \frac{1}{h^2} \Phi\left(\frac{\mathbf{x}_i - \mathbf{x}}{h}\right), \tag{3}$$

where $k$ is the number of neighbors centered in the vector, $x$, $p_h(\mathbf{x})$ denotes the probability density of $\mathbf{x}$, while $h$ is the diameter of the window that helps to satisfy the approximation of geodesic and Euclidean distances, and $\Phi$ represents the window function. This could be a rectangular function or a uniform distribution, similar to a window of size $h$. This function is also called the density estimation function. In addition, PR-Isomap alters the graph weights of a portion of each point's k-nearest neighbors with respect to the pairwise distances on the surface of the manifold, $\mathcal{M}$, that meet the requirements for the PR window. The set of a PR-limited points $\gamma\left(\mathbf{x}_i\right), \gamma\left(\mathbf{x}_i\right) \in \mathcal{M}$ are considered for projection as neighboring elements. Suppose that the point $\gamma\left(\mathbf{x}_j\right)$, where $\gamma\left(\mathbf{x}_i\right) \in \mathcal{N}_{k,h}\left(\gamma\left(\mathbf{x}_j\right)\right)$ is the closest matching point that can be used for measuring the distance $D_{g,\gamma}\left(\mathbf{x}_i, \mathbf{x}_j\right)$, i.e.,

$$\min_{\gamma(\mathbf{x}_k) \in \mathcal{N}_{k,h}(\gamma(\mathbf{x}_k))} D_{g,\gamma}\left(\mathbf{x}_i, \mathbf{x}_j\right). \tag{4}$$

Using the MDS generalized optimization, Appendix A:

$$\min_{Y} \quad \|D_{g,\gamma}(X) - D_{g,\gamma}(Y)\|_F^2. \tag{5}$$
$$\text{subject to} \quad \gamma\left(\mathbf{x}_k\right) \in \mathcal{N}_{k,h}\left(\gamma\left(\mathbf{x}_k\right)\right)$$

**Definition 2 (Density-based Embedding)** *Consider a geometric graph based on a fixed set of points* $\mathbf{x}_1, \ldots, \mathbf{x}_2 \in \mathbb{R}^d$. *Let $h$ be a real number defined for* $\mathcal{N}_{k,h}\left(\gamma\left(\mathbf{x}_k\right)\right)$ *such that* $D_{g,\gamma}\left(\mathbf{x}_i, \mathbf{x}_j\right) \leq h$ *implies that $\mathbf{x}_i$ is connected to $\mathbf{x}_j$ on the graph.*

Using Definition 2, if we rewrite MDS distance equation instead of uniform neighboring points on the HD and the LD space, but with a difference in pairwise distances in the HD space as a constraint:

$$\min_{Y} \quad \|D_{g,\gamma}(X) - D_{g,\gamma}(Y)\|_F^2 \tag{6}$$
$$\text{subject to} \quad D_{g,\gamma}\left(\mathbf{x}_i, \mathbf{x}_j\right) \leq h$$

where $D_{g,\gamma}(X)$ redefines as any pairwise distances on the surface of the HD manifold, $\mathcal{M}$, while the $\mathcal{SP}$ distance using an unweighted PR-based k-nearest neighbor graph, $p_h$. Modified Dijkstra's algorithm (Appendix B) calculates the shortest path and PR-Isomap under $\mathcal{N}_{k,h}\left(\gamma\left(\mathbf{x}_k\right)\right)$, where PR-window constraints the k-nearest neighbor path.

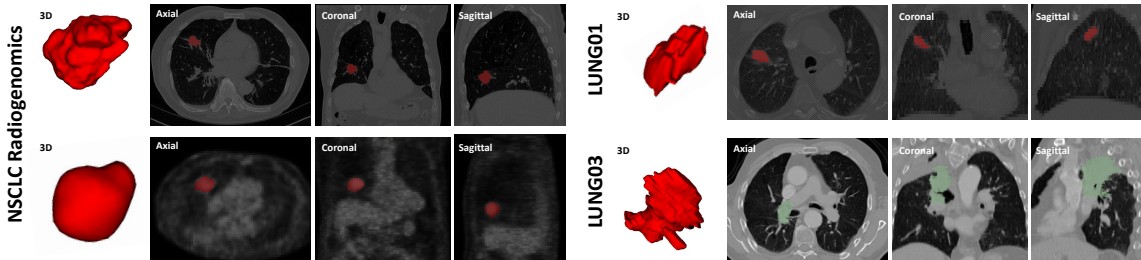

Figure 2: Visualizations of the CT/PET images with tumors in the field of view for each dataset. Tumors' morphology and shape along with other visual features affect the decision of an oncologist regarding treatment planning.

### 3.2. Treatment outcome prediction

We extracted HD radiomic features from image intensity, texture, and contextual information within the intratumor that can be seen with an oncologist, more details are in Appendix E. These HD radiomics were then transformed into an LD space while preserving maximum information. To assess the predictive performance of LD radiomics for overall survival (OS), we conducted binary classifications for treatment failure prediction. We also compared our approach to other commonly used unsupervised dimensionality reduction (DR) methods, including PR-Isomap, using multiple lung cancer datasets with diverse imaging parameters to ensure robust performance. The following techniques are remained:

1) We used various frequently used DR models to reduce HD radiomics as baselines such as the standard Isomap, tSNE, PCA, and PHATE models.

2) We conducted binary classification to investigate the power of prediction using LD radiomics of a machine learning method, random forest. We performed 10-fold cross-validation and fixed hyperparameters to ensure the quality of comparison.

3) The CPH model is used to predict OS. Kaplan Meier survival curves were also used to distinguish between high- and low- risk patients with respect to median hazard.

### 4. Results and discussion

### 4.1. Study Data

**NSCLC Radiogenomics (Multimodal PET/CT):** the NSCLC-Radiogenomics dataset Bakr et al. (2017); Gevaert et al. (2012); Clark et al. (2013) is used to provide images of 211 patients; we analyzed a sub-cohort of 130 patients for whom information on multimodal imaging was available. These provided a dataset of 57836 CT images which had corresponding PET images. We used our annotations of the tumors as observed on the medical images using maps of tumors in the CT scans and from the PET scans.

**NSCLC-Radiomics (LUNG01):** The second dataset contains images from 422 NSCLC patients. The patients' pretreatment was conducted using CT scans, and manual delineation by a radiation oncologist of the 3D volume of the total tumor volume with additive clinical outcome data Aerts et al. (2017, 2014).

**NSCLC-Radiomics (LUNG03):** The third dataset contains 88 NSCLC CT scans and

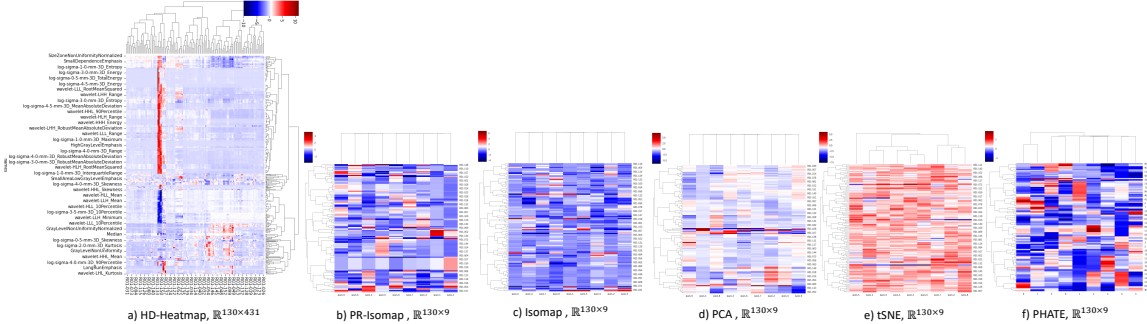

Figure 3: Visualization of the data dimensionality using heatmaps of CT radiomics of NSCLC Radiogenomics dataset using different embedding methods.

Table 1: Predictive power via random forest for three different datasets, overall of 640 cases of NSCLC.

| Datasets | Number of cases | Modality | PR-Isomap (Ours) | Isomap | PCA | tSNE | PHATE |
|---|---|---|---|---|---|---|---|
| **NSCLC Radiogenomics** | 130 | CT | **78.5 (±4.4)** | 76.9(±5.8) | 73.8 (±7.6) | 76.9(±4.1) | 76.2(±7.3) |
| | | PET | **78.5 (±5.1)** | 78.5(±6.3) | 73.2 (±5.6) | 76.9(±5.3) | 75.3(±9.9) |
| **LUNG03** | 88 | CT | **61.4 (±11.4)** | 52.7(±14.8) | 49.6 (±15.1) | 40.3(±15.5) | 45.5(±14.4) |
| **LUNG01** | 422 | CT | **88.4 (±1.4)** | 87.7(±2.8) | 86.9 (±2.5) | 87.7(±3.1) | 88.3 (±0.6) |

mRNA expression Aerts et al. (2017, 2014).

A board-certified thoracic radiologist manually segmented the tumor area using the ITK-SNAP software Yushkevich et al. (2006) (version 3.6.0) (Figure 2). Then HD radiomic features Wilson and Devaraj (2017); Van Griethuysen et al. (2017) were extracted from the images.

### 4.2. Experimental setup

**Baselines**. We compare PR density isometric mapping with four baselines, PCA Jolliffe (2011), t-SNE Van der Maaten and Hinton (2008), standard Isomap Tenenbaum et al. (2000), and PHATE Moon et al. (2019).

To visualize the LD embedded data in 3D, we use UMAP McInnes et al. (2018) to preserve the data's global structure.

**Tasks.** We employed three datasets to confirm the efficacy of PR-Isomap: NSCLC Radiogenomics Bakr et al. (2017); Gevaert et al. (2012); Clark et al. (2013), LUNG01 Aerts et al. (2017, 2014), and LUNG03 Aerts et al. (2017, 2014). More experimental details are presented in Appendix C.

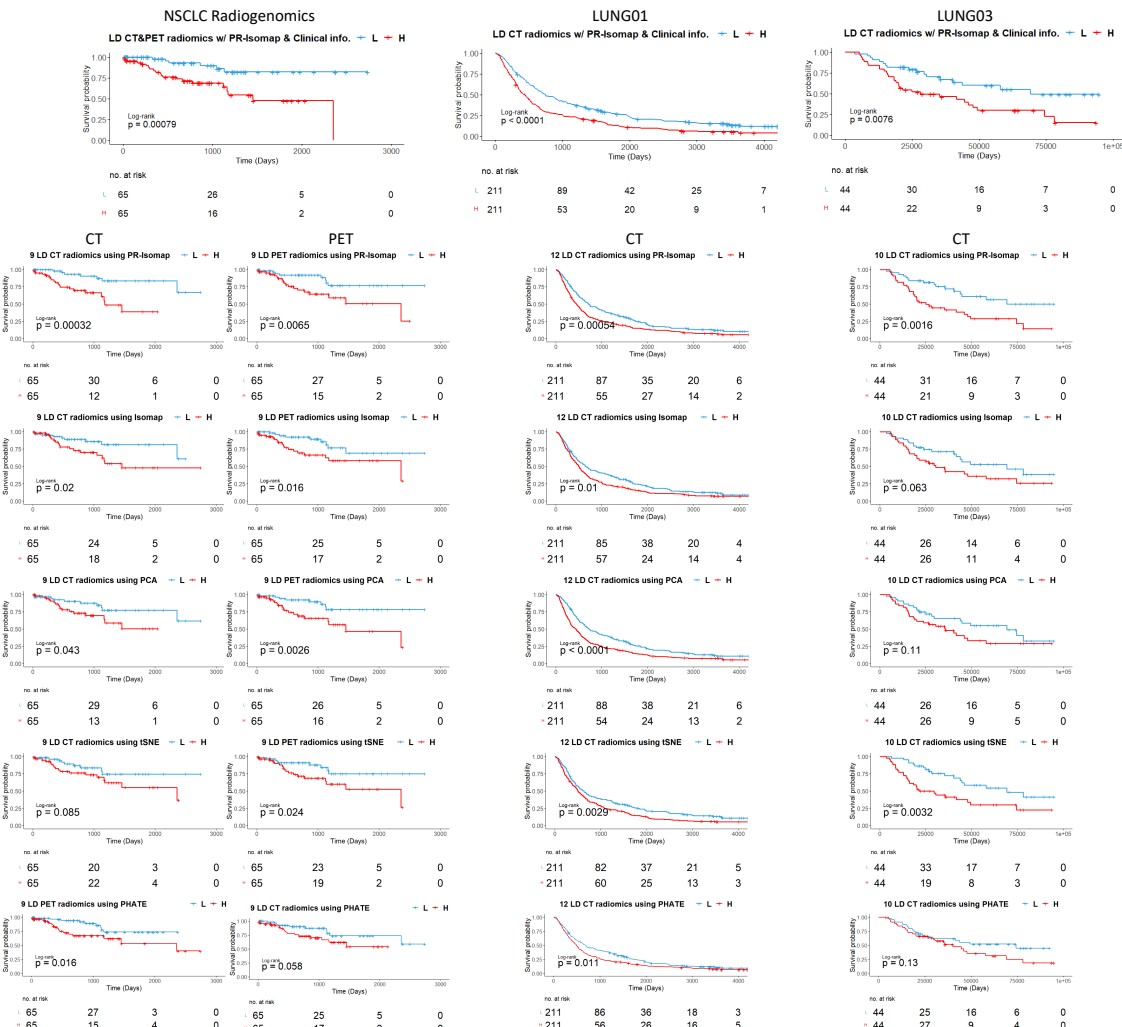

Figure 4: **LD Multivariant Survival prediction.** Density isometric mapping exhibits impressive discriminatory power in distinguishing between high-risk and low-risk patients based on the median of diverse imaging biomarkers.

## 4.3. LD Multimodal Imaging Biomarkers

Classification evaluation and outcome prediction for the patients screening or under treatment is a popular way of showing the predictability of LD radiomics. A supervised algorithm, random forest classifier, was used to perform binary classification of patients' treatment responses to assess the prediction power of LD radiomics generated by the baselines. Table 1 reports the quantitative accuracy produced by PR-Isomap, Isomap, tSNE, PCA, and PHATE. PR-Isomap produces better accuracy than other baseline techniques. Thus, PR-Isomap can be used to reduce HD radiomics and facilitate patient outcomes predictions.

### 4.4. Cox Proportional Hazard Model to Predict Survival

In Cox modeling of OS on the multimodal PET/CT dataset, the C-statistic of CT and PET LD radiomics were 0.68 and 0.67 (95% CI), with the long-rank likelihood test p-value of $< 0.005$, and 0.007, respectively. This result generated by PR-Isomap DR reduction outperformed other DR methods used for the CT biomarker analysis while maintaining the separation of high- and low- risk patients (Fig. 4). C-statistics of original Isomap while having LD CT and PET radiomics yield 0.66 (p-value = 0.02) and 0.67 (p-value = 0.02), then the results of the CT biomarkers considered to be the second-best accuracy after our proposed method. Despite slightly higher C-statistics of PHATE for the NSCLC Radiogenomics dataset, 0.70 (0.06) for CT and 0.72 (0.02) for PET, the separation of high- and low- risk patients under treatment was noticeably lower than other approaches (Fig. 4). In PET imaging markers, PCA generates the highest C statistic, which is inconsistent with binary survival prediction. the lowest C statistic belonged to CT-PCA and PET-tSNE radiomics yielding 0.63 (0.04) and 0.6 (0.02), respectively. A model including the CT-tSNE radiomic avatars resulted in C-statistics of 0.64 with a p-value of 0.09. For both Lung01 and Lung03 datasets, PR-Isomap generated LD radiomics alone yielded the highest C-Statistics of 0.59 ($< 0.0005$) and 0.66 (0.002), respectively. Also, the Kaplan-Meier survival curve indicates higher separation of the high- and low-risk patients based on the median of the hazard (Fig. 4). The Cox models of OS all datasets using PR-Isomap radiomic avatars had statistically significant separation of the Kaplan Meier curves for patients above versus below the median hazard compared to other DR methods.

### 4.5. Discussion

Prediction of treatment response using HD radiomics to mimic an oncologist for specific treatments for cancer patients using PR density isometric mapping showed considerable strength compared to other commonly used techniques. PR-Isomap still suffers from the same weaknesses as the original Isomap despite homogeneity constraint. This is due to the nature of the data, which has been processed and may be improved using different transformations to overcome the nonuniform structure of the data points on the manifold.

### 5. Conclusion

We propose predicting treatment response for NSCLC patients by looking at HD multimodal imaging data. The density-based isometric mapping can be used to reduce the dimensionality of data by projecting HD data to lower dimensional space. We use the Parzen–Rosenblatt (PR) window to add a constrain to the initial isometrically mapping optimization maintaining more uniformity in the data projection. We applied density-based isometric mapping to create LD projection of three NSCLC datasets containing multimodal PET/CT and measured the classification accuracy and the predicted response of the LD radiomics, to calculate the survival of the patients under specific treatment. The density-based isometric mapping produces better outcomes than several popular unsupervised DR methods to imitate an oncologist.

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

## Appendix A. MDS

Our data points can be shown as a combination of multiple vectorized data points, $X = [\mathbf{x}_1, \mathbf{x}_2, \ldots, \mathbf{x}_n] \in \mathbb{R}^{d \times n}$, where $\{\mathbf{x}_i \in \mathbb{R}^d\}_{i=1}^n$. We assume that there is a finite data point drawn from our X. We can build a geometric graph $G = (V; E)$ that has our data points as vertices and connects vertices that are close to each other. We construct a k-nearest neighbor graph for the $\mathbf{x}_i$ connected to $\mathbf{x}_j$ correspondingly and for $\mathbf{x}_j$ to $\mathbf{x}_i$, show them by $(\mathbf{x}_i)$ and $\gamma(\mathbf{x}_j), \gamma(\mathbf{x}_i), \gamma(\mathbf{x}_j) \in \mathcal{M}$. $\gamma(.)$ represents a vector point on the manifold. The

length of the path is defined by summing the edge weights along the path on the manifold. The shortest path ($\mathcal{SP}$) distance $D_{\mathcal{SP}}(\gamma(\mathbf{x}_i), \gamma(\mathbf{x}_j))$ between two vertices $\gamma(\mathbf{x}_i), \gamma(\mathbf{x}_j)$ and defines based on the length of the $\mathcal{SP}$ connecting them. $g(\cdot)$ is a positive continuous function for $X$ in the given path $\gamma(\cdot)$ connecting $\mathbf{x}_i$ and $\mathbf{x}_j$, which is parameterized by $g$-length of the path as

$$D_{g,\gamma}(t) = \int_\gamma g(\gamma(t)) \mid \gamma'(t) \quad dt, \tag{7}$$

where $\gamma'(\cdot)$ defines a small element on the manifold, also known as the line integral along the path $\gamma(\cdot)$ with respect to $g(\cdot)$. $D_{g,\gamma}$ denotes as distance on the manifold through the shortest path. The g-geodesic path connecting $\mathbf{x}_i$ and $\mathbf{x}_j$ is the path with minimized g-length Alamgir and Von Luxburg (2012). The shortest path is used for the Isomap algorithm to find the path on the HD manifold for connecting two points. We assume that the LD embedding of the training data with respect to the vectorized data points of the observation can be shown as $Y = [\mathbf{y}_1, \mathbf{y}_2, \ldots, \mathbf{y}_n] \in \mathbb{R}^{p \times n}$, where $\{\mathbf{y}_i \in \mathbb{R}^p\}_{i=1}^n$, and $p \leq d$, desirably and usually $p \ll d$. The MDS method aims to project similar points of the data close to each other and dissimilar points as far as possible. This is followed by the equation below:

$$\min_Y \|D_{g,\gamma}(X) - D_{g,\gamma}(Y)\|_F^2, \tag{8}$$

where $D_{g,\gamma}(\cdot)$ is the shortest path distance from equation (1). $D_{g,\gamma}(X)$ and $D_{g,\gamma}(Y)$ are any pairwise distances of the data in HD space and in the subspace and can be written by similarity $X^T X$ and $Y^T Y$ in the projected space, respectively. In MDS, the solution for this problem is $Y = \Delta^{\frac{1}{2}} V^T$, where $V^T$ and $\Delta$ are the eigenvector and eigenvalue matrices of the input data, $X$.

## Appendix B. Modified Algorithm

Density-based isometric embedding modifies Dijkstra's algorithm is presented in Algorithm 1:

## Appendix C. Experiment details

**Implementation Details.** To apply PCA and the standard t-SNE, we use the implementation from Minka (2000), and Hinton and van der Maaten (2008), respectively. To apply UMAP for LD data visualization, we apply the implementation from McInnes et al. (2018). For standard Isomap, we use Tenenbaum et al. (2000) implemented by Jake Vanderplas (2011) [1]. We implement PR-Isomap based on the standard implementation of Isomap by restricting k-nearest neighbors and the shortest path on the manifold and updating it with the PR constraint.

**Hyperparameters.** For PCA and UMAP, we use the default hyperparameters, while we

---

1. Original implementation of `isometric mapping` https://scikit-learn.org/0.18/auto_examples/manifold

---

**Algorithm 1:** PR-Isometric Mapping. Dijkstra's Algorithm

---

**Data:** $\{\mathbf{x}_1, \mathbf{x}_1, \ldots, \mathbf{x}_n\} \in \mathbb{R}^D$.

**Result:** $\{\mathbf{y}_1, \mathbf{y}_1, \ldots, \mathbf{y}_n\} \in \mathbb{R}^d$.

**Estimate:** MDS on $D$- Geodesic distance pairwise point via shortest path & neighborhood graph dissimilarity matrix $D$. $D \in \mathbb{R}^{n \times n}(D_{ii} = 0, D_{ij} \geq 0)$, $K = -\frac{1}{2}HDH$

$K = -\frac{1}{2}HDH$ centering matrix of $H$, $H = I - \frac{1}{n}11^T$

$Y = \Delta^{\frac{1}{2}} V^T$

**Constrained Dijkstra's Algorithm**

Form a weighted undirected graph $\mathcal{G}_e = (\nu_e, E_e, w_e)$ for q-nearest neighbor, $q \leq k$.

For $\forall$ vertex $v$, $v \in \mathcal{G} : D_\gamma[v] \leftarrow \infty$, $parent[v] \leftarrow []$, $D_\gamma[s]] \leftarrow 0$,

$Q := \forall$ nodes,$n$, $n \in \mathcal{G}$

**while** $Q \neq []$ **do**

> $u \leftarrow v \in Q$ & $\min D_\gamma[u]$
>
> remove $u$ from Q
>
> **for** $\forall \mathcal{N}_q[u, v] \in Q$ **do**
>
> > **if** $D_\gamma[u, v] < h$, $u, v \in \mathcal{N}_{k,h}$ **then**
> >
> > > $alt \leftarrow D_\gamma[u] + \mathcal{G}_e edge(u, v)$
> > >
> > > **if** $alt_i D_\gamma[v]$ **then**
> > >
> > > > $D_\gamma[v] \leftarrow alt$
> > > >
> > > > $parent[v] \leftarrow u$
> > >
> > > **end**
> >
> > **end**
>
> **end**

**end**

---

changed the tSNE iteration number to 500 and a random state of 3. For the binary classifiers, we used untuned classification models but we feezed the hyperparameters throughout the comparison for different DR methods and all models to keep the integrity of the analysis. In Random Forest, we used 20 estimators, a maximum depth of the forest with four, and a random state of 45. In the Naïve Bayes, logistic regression, and support vector machine, we use the default hyperparameters. The training follows the standard cross-validation score with k = 8. For PR-Isomap, $\Phi$ was a window function. h in the PR function was 16, and 10 for CT and PET radiomics for converting LD features.

## Appendix D. Computational Complexity

PR-Isomap changes the complexity of Dijkstra's algorithm on the new graph (see Algorithm 1), $G_e$, is $O(log(N_e)E_e)$, $N_e$ and $E_e$ are the number of nodes and edges in $G_e$, respectively. For a $q$-nearest-neighbor graph with the parameter $q$, we have $qN/2$ edges, resulting in $N_e = qN/2$ nodes in $G_e$. Connecting each node to all $q$ neighbors, edges in $G_e$ in the worst

case scenario is $E_e = ((qN/2)q)/2$, leading to a complexity of $O(log(qN/2)((qN/2)q)/2) = O((q^2N/4)log(qN/2))$.

## Appendix E. HD Medical Imaging Features

To extract radiomic features, the following steps must be taken: image acquisition, selection of the region of interest (ROI) (usually a segmented tumor), and the application of various imaging parameters (including non-contrast and contrast images, different convolutional kernels, and varying slice thicknesses). A total of 862 features were extracted from these images, 431 features from each imaging modality (Figure 4), divided into nine categories: first-order statistics (FO), shape-based expression (SB), gray level co-occurrence matrix (GLCM), gray level dependence matrix (GLDM), gray level run length matrix (GLRLM), gray level size zone matrix (GLSZM), neighboring gray-tone difference matrix (NGTDM), Laplacian of Gaussian (LOG), and three-layer filtering wavelet Wilson and Devaraj (2017); Van Griethuysen et al. (2017).

