# OpenReview forum: "Emulating Oncologists' Gaze for Predicting Treatment Response through Multimodal Imaging"
_NeurIPS.cc/2023/Workshop/Gaze_Meets_ML — Submitted to Gaze Meets ML 2023_

### Official Review · Reviewer_czFG · 2023-10-15
**A poorly written paper proposing a "new" non-linear dimensionality reduction technique**

**Rating:** 2
**Confidence:** 5

**Review:**

This paper has gaze in the title, but otherwise makes no mention of gaze. It is not aligned with the workshop topic "Gaze Meets ML".

Quality

The paper is poorly written. In the abstract the authors state "We achieved prediction accuracy comparable to newly and commonly established methods". In the discussion, they state "Prediction of treatment response using HD radiomics to mimic an oncologist for specific treatments for cancer patients using PR density isometric mapping showed considerable strength compared to other commonly used techniques."  Which is it? Is the proposed method comparable or better?

Table 1 shows the accuracy (which the authors bizarrely refer to as "quantitative accuracy"-- is there a qualitative accuracy??) of a random forest binary classifier using the different dimensionality reduction techniques. For the NSCLC Radiogenomics and LUNG01 datasets, the accuracy of PR-Isomap is within the standard error of almost all other techniques. For the LUNG03 dataset, all methods perform poorly (40-61%) and have a large (11-15%) standard error. I conclude that the PR-Isomap is no better nor worse than the other techniques.

The tSNE and UMAP methods used the default settings for perplexity/neighborhood. These methods could have been significantly better with such optimization. This makes it an unfair comparison.

There are too many spelling and grammatical mistakes. Briefly:
* Introduction: "The abundance of radiomics, high-dimensional (HD), impedes the models to accurately predict the outcome." The sentence is gibberish. Perhaps it would be clearer to say "The high-dimensionality of radiomics impedes the development of accurate models."
* Introduction: "Aerts et al. (2017, 2014), which are focus on computed tomography". The phrase doesn't make sense.
* Introduction: "the overall strategy of the proposes system". Change to "proposed system".
* Throughout: The "Parzen-Rosenblatt constrain". Change to "constraint".

Too many unnecessary abbreviations.
* Section 3.2: "The CPH model is used to predict OS."  Spell out Cox Proportional Hazard and Outcome Survival.

Clarity

Figure 4 shows the multi-year survival predictions. The figure has too many graphs. It should be reformatted into multiple figures for better clarity. The log-ranks should be converted to a simple table to allow the reader an easier way to compare the techniques.

The Discussion and Conclusion sections are one paragraph each. In the Discussion, the authors first sentence states that the PR-Isomap "showed considerable strength", but the second sentence states "PR-Isomap still suffers from the same weakness of the original Isomap". These two sentences seem to contradict.

Originality

The paper is not original. The first line of the one paragraph conclusion says: "We propose predicting treatment response for NSCLC patients by looking at HD multimodal imaging data." Using dimensionality reduction is standard practice in all machine learning models. Many studies have shown benefit from combining multimodal imaging in the prediction of survival.

Significance of the Work

None.

Pros
The use of the classification model and survivability model was appropriate.


Cons
* This paper does not analyze gaze data. It doesn't fit the purpose of the workshop.
* Not original
* Did not look at fine tuning the hyperparameters of the alternate methods (tSNE, UMAP)
* Multiple grammatical and spelling mistakes
* Many unnecessary abbreviations
* On page 4, missing Figure reference (it says Algorithm 1 and Figure ??)

---

### Official Review · Reviewer_xQzw · 2023-10-23
**Emulating Oncologists' Gaze for Predicting Treatment Response through Multimodal Imaging**

**Rating:** 9
**Confidence:** 5

**Review:**

This study aims to imitate oncologists' decisions about the response to therapy by looking at lung tumours in different imaging modalities. The authors did a great job of using three datasets to evaluate the performance of their model, however, it would be interesting to see if more data (such as 5-10 datasets) can improve the overall performance of their model. Similarly, since lung tumour features differed from one patient to another, the authors should state how well they hope their model would perform in terms of patient diversity. The authors also did a good job of avoiding bias in their comparison while achieving a somewhat good accuracy. Collectively, this is a great study that, if expanded on by extensive patient datasets and validation, might contribute to predicting patient outcomes in response to therapy.

---

### Official Review · Reviewer_LLud · 2023-10-23
**This is a paper that introduces a dimensionality reduction to use radiomics in prediction of cancer treatment response.**

**Rating:** 4
**Confidence:** 2

**Review:**

The paper proposes a dimensionality reduction for radiomics obtained from CT and PET to be used for prediction of treatment response in cancer patients. As a paper on treatment prediction, it seems to be a good paper. However, I fail to see any relevance between the paper and gaze tracking. Gaze tracking only appears in the title of the paper.

---

### Meta-Review · Area_Chair_GfCi · 2023-10-26

**Recommendation:** Reject
**Confidence:** 5

**Metareview:**

The paper proposes a dimensionality reduction for radiomics obtained from CT and PET to be used for prediction of treatment response in cancer patients, which appears to be a reasonable paper on treatment prediction. However, I agree with the reviewers that there is no apparent relevance between this paper and gaze tracking apart from mention of gaze in the title of the paper. Recommend reject for lack of relevance to workshop topic.

---

### Decision · Program_Chairs · 2023-10-26

Reject